# Autoregressive Perturbations for Data Poisoning

**Pedro Sandoval-Segura**[1][*] **Vasu Singla**[1][*] **Jonas Geiping**[1] **Micah Goldblum**[2]
**Tom Goldstein**[1] **David W. Jacobs**[1]
[1]University of Maryland   [2]New York University
{psando, vsingla, jgeiping, tomg, dwj}@umd.edu   goldblum@nyu.edu

## Abstract

The prevalence of data scraping from social media as a means to obtain datasets has led to growing concerns regarding unauthorized use of data. Data poisoning attacks have been proposed as a bulwark against scraping, as they make data "unlearnable" by adding small, imperceptible perturbations. Unfortunately, existing methods require knowledge of both the target architecture and the complete dataset so that a surrogate network can be trained, the parameters of which are used to generate the attack. In this work, we introduce autoregressive (AR) poisoning, a method that can generate poisoned data without access to the broader dataset. The proposed AR perturbations are generic, can be applied across different datasets, and can poison different architectures. Compared to existing unlearnable methods, our AR poisons are more resistant against common defenses such as adversarial training and strong data augmentations. Our analysis further provides insight into what makes an effective data poison.

## 1 Introduction

Increasingly large datasets are being used to train state-of-the-art neural networks [24, 26, 25]. But collecting enormous datasets through web scraping makes it intractable for a human to review samples in a meaningful way or to obtain consent from relevant parties [3]. In fact, companies have already trained commercial facial recognition systems using personal data collected from media platforms [15]. To prevent the further exploitation of online data for unauthorized or illegal purposes, imperceptible, adversarial modifications to images can be crafted to cause erroneous output for a neural network trained on the modified data [12]. This crafting of malicious perturbations for the purpose of interfering with model training is known as data poisoning.

In this work, we focus on poisoning data to induce poor performance for a network trained on the perturbed data. This kind of indiscriminate poisoning, which seeks to damage average model performance, is often referred to as an availability attack [1, 2, 40, 18, 9, 10]. Because we assume the data is hosted on a central server controlled by the poisoner, the poisoner is allowed to perturb the entire dataset, or a large portion of it. Throughout this work, unless stated otherwise, poisoning refers to the perturbing of every image in the training dataset. This makes the creation of unlearnable data different from other poisoning methods, such as backdoor [5, 13] and targeted poisoning attacks [28, 43].

We introduce autoregressive (AR) data poisoning for degrading overall performance of neural networks on clean data. The perturbations that we additively apply to clean data are generated by AR processes that are data and architecture-independent. An AR($p$) process is a Markov chain, where each new element is a linear combination of $p$ previous ones, plus noise. This means AR perturbations are cheap to generate, not requiring any optimization or backpropagation through network parameters. AR perturbations are generic; the same set of AR processes can be re-used to

---

[*]Authors contributed equally.

36th Conference on Neural Information Processing Systems (NeurIPS 2022).

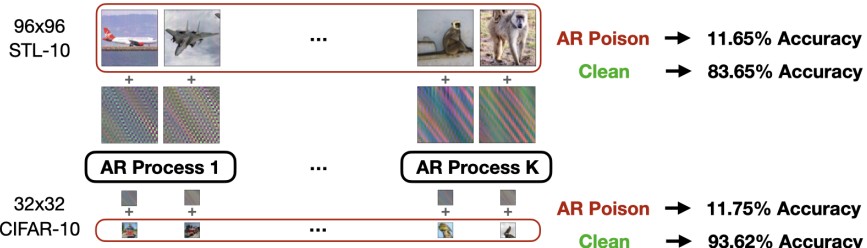

Figure 1: The same set of AR processes can generate perturbations for different kinds of data without access to a surrogate network or other images in a dataset. A ResNet-18 trained on these AR poisons is unable to generalize to clean test data, even with the help of strong data augmentations.

generate diverse perturbations for different image sizes and new datasets, unlike other poisoning methods which need to train a surrogate network on the target dataset before crafting perturbations.

Our method also provides new insight into why data poisoning works. We work on top of the result that effective poisons are typically easy to learn [27] and construct AR perturbations which are separable by a manually-specified CNN. Working under the intuition that highly separable perturbations should be easily learned, we use the manual specification of parameters as a way of demonstrating that our AR perturbations are easily separable. Our manually-specified CNN makes use of what we call AR filters, which are attuned to detect noise from a specific AR process. AR poisoning's effectiveness is competitive or better than error-maximizing, error-minimizing, and random noise poisoning across a range of architectures, datasets, and common defenses. AR poisoning represents a paradigm shift for what a successful indiscriminate poisoning attack looks like, and raises the question of whether strong indiscriminate poisons need to be generated by surrogate networks for a given dataset.

## 2 Background & Related Work

**Error-minimizing and Error-maximizing Noise.** To conduct poisoning attacks on neural networks, recent works have modified data to explicitly cause gradient vanishing [31] or to *minimize* the loss with respect to the input image [18]. Images perturbed with error-minimizing noises are a surprisingly good data poisoning attack. A ResNet-18 (RN-18) trained on a CIFAR-10 [20] sample-wise error-minimizing poison achieves $19.9\%$ final test accuracy, while the class-wise variant achieves $16.4\%$ final test accuracy after 60 epochs of training [18]. More recently, strong adversarial attacks, which perturb clean data by *maximizing* the loss with respect to the input image, have been shown to be the most successful approach thus far [10]. An error-maximizing poison can poison a network to achieve $6.25\%$ test accuracy on CIFAR-10. But both error-minimizing and error-maximizing poisons require a surrogate network, from which perturbations are optimized. The optimization can be expensive. For example, crafting the main CIFAR-10 poison from [10] takes roughly 6 hours on 4 GPUs. In contrast, our AR perturbations do not require access to network parameters and can be generated quickly, without the need for backpropagation or a GPU. We provide a technical overview of error-minimizing and error-maximizing perturbations in Section 3.1.

**Random Noise.** Given their simplicity, random noises for data poisoning have been explored as necessary baselines for indiscriminate poisoning. If random noise, constrained by an $\ell_\infty$ norm, is applied sample-wise to every image in CIFAR-10, a RN-18 trained on this poison can still generalize to the test set, with ~90% accuracy [10, 18]. But if the noise is applied class-wise, where every image of a class is modified with an identical additive perturbation, then a RN-18 trained on this CIFAR-10 poison will achieve around chance accuracy; *i.e.* ~10% [39, 18, 27]. The random perturbations of [39] consist of a fixed number of uniform patch regions, and are nearly identical to the class-wise poison, called "Regions-16," from [27]. All the random noises that we consider are class-wise, and we confirm they work well in a standard training setup using a RN-18, but their performance varies across architectures and they are rendered ineffective against strong data augmentations like Cutout [7], CutMix [41], and Mixup [42]. Conversely, our AR poisons degrade test performance more than error-maximizing, error-minimizing, and random poisons on almost every architecture. We show that AR perturbations are effective against strong data augmentations and can even mitigate some effects of adversarial training.

**Understanding Poisoning.** A few works have explored properties that make for effective poisons. For example, [27] find that poisons which are learned quickly have a more harmful effect on the poison-trained network, suggesting that the more quickly perturbations help minimize the training loss, the more effective the poison is. [39] perform a related experiment where they use a single linear layer, train on perturbations from a variety of poisoning methods, and demonstrate that they can discriminate whether a perturbation is error-minimizing or error-maximizing with high accuracy. We make use of ideas from both papers, designing AR perturbations that are provably separable and Markovian in local regions.

**Other Related Work.** Several works have also focused on variants of "unlearnable" poisoning attacks. [9] propose to employ gradient alignment [11] to generate poisons. But their method is computationally expensive; it requires a surrogate model to solve a bi-level objective. [40] propose generation of an unlearnable dataset using neural tangent kernels. Their method also requires training a surrogate model, takes a long time to generate, and does not scale easily to large datasets. In contrast, our approach is simple and does not require surrogate models. [23] propose an invertible transformation to control learnability of a dataset for authorized users, while ensuring the data remains unlearnable for other users. [35] showed that data poisoning methods can be broken using adversarial training. [30] and [37] propose variants of error-minimizing noise to defend against adversarial training. Our AR poisons do not focus on adversarial training. While adversarial training remains a strong defense, our AR poisons show competitive performance. We discuss adversarial training in detail in Section 4.3.2. A thorough overview of data poisoning methods, including those that do not perturb the entire training dataset, can be found in [12].

# 3 Autoregressive Noises for Poisoning

## 3.1 Problem Statement

We formulate the problem of creating a clean-label poison in the context of image classification with DNNs, following [18]. For a $K$-class classification task, we denote the clean training and test datasets as $\mathcal{D}_c$ and $\mathcal{D}_t$, respectively. We assume $\mathcal{D}_c, \mathcal{D}_t \sim \mathcal{D}$. We let $f_\theta$ represent a classification DNN with parameters $\theta$. The goal is to perturb $\mathcal{D}_c$ into a poisoned set $\mathcal{D}_p$ such that when DNNs are trained on $\mathcal{D}_p$, they perform poorly on test set $\mathcal{D}_t$.

Suppose there are $n$ samples in the clean training set, i.e. $\mathcal{D}_c = \{(x_i, y_i)\}_{i=1}^n$ where $x_i \in \mathbb{R}^d$ are the inputs and $y_i \in \{1, ..., K\}$ are the labels. We denote the poisoned dataset as $\mathcal{D}_p = \{(x_i', y_i)\}_{i=1}^n$ where $x_i' = x_i + \delta_i$ is the poisoned version of the example $x_i \in \mathcal{D}_c$ and where $\delta_i \in \Delta \subset \mathbb{R}^d$ is the perturbation. The set of allowable perturbations, $\Delta$, is usually defined by $\|\delta\|_p < \epsilon$ where $\|\cdot\|_p$ is the $\ell_p$ norm and $\epsilon$ is set to be small enough that it does not affect the utility of the example. In this work, we use the $\ell_2$ norm to constrain the size of our perturbations for reasons we describe in Section 3.4.

Poisons are created by applying a perturbation to a clean image in either a class-wise or sample-wise manner. When a perturbation is applied class-wise, every sample of a given class is perturbed in the same way. That is, $x_i' = x_i + \delta_{y_i}$ and $\delta_{y_i} \in \Delta_C = \{\delta_1, ..., \delta_K\}$. Due to the explicit correlation between the perturbation and the true label, it should not be surprising that class-wise poisons appear to trick the model to learn the perturbation over the image content, subsequently reducing generalization to the clean test set. When a poison is applied sample-wise, every sample of the training set is perturbed independently. That is, $x_i' = x_i + \delta_i$ and $\delta_i \in \Delta_S = \{\delta_1, ..., \delta_n\}$. Because class-wise perturbations can be recovered by taking the average image of a class, these should therefore be easy to remove. Hence, we focus our study on sample-wise instead of class-wise poisons. We still compare to simple, randomly generated class-wise noises shown by [18] to further demonstrate the effectiveness of our method.

All indiscriminate poisoning aims to solve the following bi-level objective:

$$\max_{\delta \in \Delta} \mathbb{E}_{(x,y) \sim \mathcal{D}_t} \left[ \mathcal{L}(f(x), y; \theta(\delta)) \right] \tag{1}$$

$$\theta(\delta) = \arg\min_\theta \mathbb{E}_{(x_i, y_i) \sim \mathcal{D}_c} \left[ \mathcal{L}(f(x_i + \delta_i), y_i; \theta) \right] \tag{2}$$

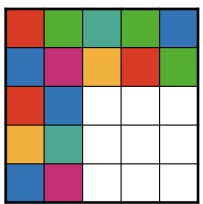 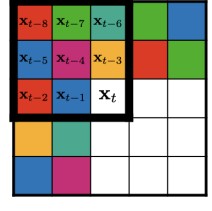 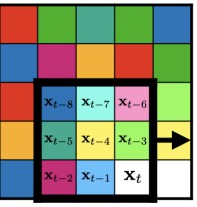 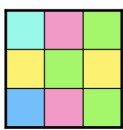

| 1. Gaussian start signal | 2. Use AR process in window to compute next value | 3. Sliding window | 4. Crop & scale |

Figure 2: To generate 2D noise for a single channel using a $3 \times 3$ sliding window, we first sample Gaussian noise for the first 2 columns and rows. Then, we use an AR(8) process within a sliding window to compute the next value. Finally, we crop and scale the perturbation to be within our $\ell_2$ constraint.

Eq. 2 describes the process of training a network on poisoned data; i.e. $x_i$ perturbed by $\delta_i$. Eq. 1 states that the poisoned network should maximize the loss, and thus perform poorly, on clean test data.

Different approaches have been proposed to construct $\delta_i$. Both error-maximizing [10] and error-minimizing [18] poisoning approaches use a surrogate network, trained on clean training data, to optimize perturbations. We denote surrogate network parameters as $\theta^*$. Error-maximizing poisoning [10] proposes constructing $\delta_i$ that maximize the loss of the surrogate network on clean training data:

$$\max_{\delta \in \Delta} \mathbb{E}_{(x_i, y_i) \sim \mathcal{D}_c} \left[ \mathcal{L}(f(x_i + \delta_i), y_i; \theta^*) \right] \tag{3}$$

whereas error-minimizing poisoning [18] solve the following objective to construct $\delta_i$ that minimize the loss of the surrogate network on clean training data:

$$\min_{\delta \in \Delta} \mathbb{E}_{(x_i, y_i) \sim \mathcal{D}_c} \left[ \mathcal{L}(f(x_i + \delta_i), y_i; \theta^*) \right] \tag{4}$$

In both error-maximizing and error-minimizing poisoning, the adversary intends for a network, $f$, trained on the poison to perform poorly on the test distribution $D_t$, from which $D_c$ was also sampled. But the way in which both methods achieve the same goal is distinct.

## 3.2   Generating Autoregressive Noise

Autoregressive (AR) perturbations have a particularly useful structure where local regions throughout the perturbation are Markovian, exposing a linear dependence on neighboring pixels [38]. This property is critical as it allows for a particular filter to perfectly detect noise from a specific AR process, indicating the noise is simple and potentially easily learned.

We develop a sample-wise poison where clean images are perturbed using additive noise. For each $x_i$ in the clean training dataset, our algorithm crafts a $\delta_i$, where $\|\delta_i\|_2 \leq \epsilon$, so that the resulting poison image is $x_i' = x_i + \delta_i$. The novelty of our method is in how we find and use autoregressive (AR) processes to generate $\delta_i$. In the following, let $\mathbf{x}_t$ refer to the $t^{\text{th}}$ entry within a sliding window of $\delta_i$.

An autoregressive (AR) process models the conditional mean of $\mathbf{x}_t$, as a function of past observations $\mathbf{x}_{t-1}, \mathbf{x}_{t-2}, ..., \mathbf{x}_{t-p}$ in the following way:

$$\mathbf{x}_t = \beta_1 \mathbf{x}_{t-1} + \beta_2 \mathbf{x}_{t-2} + ... + \beta_p \mathbf{x}_{t-p} + \epsilon_t \tag{5}$$

where $\epsilon_t$ is an uncorrelated process with mean zero and $\beta_i$ are the AR process coefficients. For simplicity, we set $\epsilon_t = 0$ in our work. An AR process that depends on $p$ past observations is called an AR model of degree $p$, denoted AR($p$). For any AR($p$) process, we can construct a size $p + 1$ filter where the elements are $\beta_p, ..., \beta_1$ and the last entry of the filter is $-1$. This filter produces a zero response for any signal generated by the AR process with coefficients $\beta_p, ..., \beta_1$. We refer to this filter as an AR filter, the utility of which is explained in Section 3.3 and Appendix A.1.

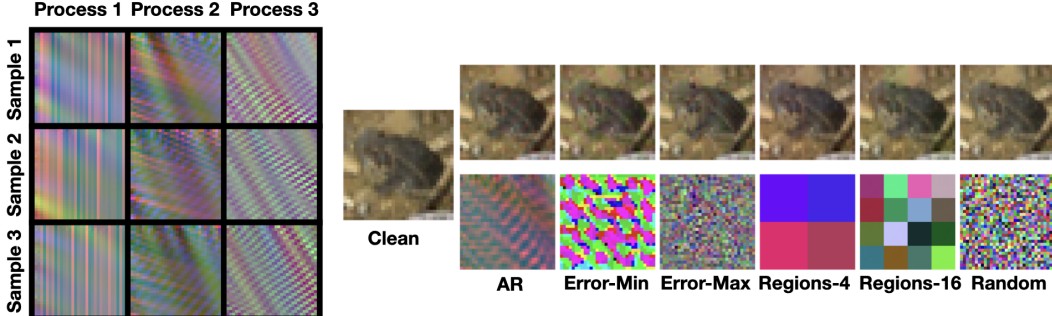

Figure 3: Left: Normalized samples of perturbations generated by 3 AR processes. Right: Poisoned images and the corresponding perturbation for a randomly selected CIFAR-10 image.

Suppose we have a $K$ class classification problem of $H \times W \times C$ dimensional images. For each class label $y_i$, we construct a set $A_{y_i}$ of AR processes, one for each of the $C$ channels. For each of the $C$ channels, we will be applying an AR process from $A_{y_i}$ inside a $V \times V$ sliding window. Naturally, using an AR process requires initial observations, so we populate the perturbation vector $\delta_i$ with Gaussian noise for the first $V - 1$ columns and rows. The $V \times V$ sliding window starts at the top left corner of $\delta_i$. Within this sliding window, we apply the AR($V^2 - 1$) process: the first $V^2 - 1$ entries in the sliding window are considered previously generates (or randomly initialized) entries in the 2D array $\delta_i$, and the $V^{\text{th}}$ entry is computed by Eq. 5. The window is slid left to right, top to bottom until the first channel of $\delta_i$ is filled. We then proceed to use the next AR($V^2 - 1$) process in $A_{y_i}$ for the remaining $C - 1$ channels. Finally, we discard the random Gaussian rows and columns used for initialization, and scale $\delta_i$ to be of size $\epsilon$ in the $\ell_2$-norm. Note that this sliding window procedure resembles that of a convolution. That is by design, and we explain why it is important in Section 3.3. A high-level overview of this algorithm is illustrated in Figure 2. Additional details are in Appendix A.3.2. While we describe our use of AR processes on $C$-channel images, our method could, in principle, be applied to data other than images. Note that these AR perturbations are fast to generate, do not require a pre-trained surrogate model, and can be generated *independently from the data.*

## 3.3 Why do Autoregressive Perturbations Work?

Perturbations that are easy to learn have been shown to be more effective at data poisoning [27]. Intuitively, a signal that is easily interpolated by a network will be quickly identified and used as a "shortcut," whereas complex and unpredictable patterns may not be learned until after a network has already extracted useful content-based features [29]. Thus, we seek imperceptible perturbations that are easy to learn. We propose a simple hypothesis: if there exists a simple CNN that can classify autoregressive signals perfectly, then these signals will be easy to learn. The signals can then be applied to clean images and serve as a shortcut for learning by commonly-used CNNs.

Autoregressive perturbations, despite looking visually complex, are actually very simple. To demonstrate their separability, we manually specify the parameters of a simple CNN that classifies AR perturbations perfectly by using AR filters. In the following, we prove AR filters satisfy an important property.

**Lemma 3.1.** *Given an AR perturbation $\delta$, generated from an AR(p) with coefficients $\beta_1, ..., \beta_p$, there exists a linear, shift invariant filter where the cross-correlation operator produces a zero response.*

We provide a proof in Appendix A.1. The construction of an AR filter that produces a zero response for any noise generated from the corresponding AR process is useful because we can construct a CNN which makes use of solely these AR filters to classify signals. That is, given any AR perturbation, the AR filter with the zero response correctly designates the AR process from which the perturbation was generated. We verify this claim in Appendix A.2 by specifying the 3-layer CNN that can perfectly classify AR perturbations.

Crucially, we are not interested in learning classes of AR signals. Rather, we are interested in how quickly a model can learn classes of clean data *perturbed* by AR signals. Nevertheless, the

characterization of our AR perturbations as easy to learn, demonstrated by the manual specification of a 3-layer CNN, is certainly an indication that, when applied to clean data, AR perturbations can serve as bait for CNNs. Our experiments will seek to answer the following question: If we perturb each sample in the training dataset with an imperceptible, yet easily learned AR perturbation, can we induce a learning "shortcut" that minimizes the training loss but prevents generalization?

### 3.4 Finding AR Process Coefficients

We generate AR processes using a random search that promotes diversity. We generate processes one-at-a-time by starting with a random Gaussian vector of coefficients. We then scale the coefficients so that they sum to one. We then append a $-1$ to the end of the coefficients to produce the associated AR filter, and convolve this filter with previously generated perturbations. We use the norms of the resulting convolution outputs as a measure of similarity between processes. If the minimum of these norms is below a cutoff $T$, then we deem the AR process too coherent with previously generated perturbations – the coefficients are discarded and we try again with a different random vector.

Once the AR process coefficients are identified for a class, we use them to produce a perturbation $\delta_i$ for each image in the class. This perturbation is scaled to be exactly of size $\epsilon$ in the $\ell_2$-norm. To level the playing field among all poisoning methods, we measure all perturbations using an $\ell_2$ norm in this work. A more detailed description of this process can be found in Appendix A.3.1.

## 4 Experiments

We demonstrate the generality of AR poisoning by creating poisons across four datasets, including different image sizes and number of classes. Notably, we use the *same* set of AR processes to poison SVHN [22], STL-10 [6], and CIFAR-10 [20] since all of these datasets are 10 class classification problems. We demonstrate that despite the victim's choice of network architecture, AR poisons can degrade a network's accuracy on clean test data. We show that while strong data augmentations are an effective defense against all poisons we consider, AR poisoning is largely resistant. Adversarial training and diluting the poison with clean data remain strong defenses, but our AR poisoning method is competitive with other poisons we consider. All experiments follow the same general pattern: we train a network on a poisoned dataset and then evaluate the trained network's performance on clean test data. A poison is effective if it can cause the trained network to have poor test accuracy on clean data, so lower numbers are better throughout our results.

**Experimental Settings.** We train a number of ResNet-18 (RN-18) [14] models on different poisons with cross-entropy loss for 100 epochs using a batch size of 128. For our optimizer, we use SGD with momentum of 0.9 and weight decay of $5 \times 10^{-4}$. We use an initial learning rate of 0.1, which decays by a factor of 10 on epoch 50. In Table 2, we use use the same settings with different network architectures.

### 4.1 Error-Max, Error-Min, and other Random Noise Poisons

SVHN [22], CIFAR-10, and CIFAR-100 [20] poisons considered in this work contain perturbations of size $\epsilon = 1$ in $\ell_2$, unless stated otherwise. For STL-10 [6], all poisons use perturbations of size $\epsilon = 3$ in $\ell_2$ due to the larger size of STL-10 images. In all cases, perturbations are normalized and scaled to be of size $\epsilon$ in $\ell_2$, are additively applied to clean data, and are subsequently clamped to be in image space. Dataset details can be found in Appendix A.4. A sampling of poison images and their corresponding normalized perturbation can be found in Figure 3 and Appendix A.8. In our results, class-wise poisons are marked with ∘ and sample-wise poisons are marked with •.

**Error-Max and Error-Min Noise.** To generate error-maximizing poisons, we use the open-source implementation of [10]. In particular, we use a 250-step $\ell_2$ PGD attack to optimize Eq. (3). To generate error-minimizing poisons, we use the open-source implementation of [18], where a 20-step $\ell_2$ PGD attack is used to optimize Eq. (4). For error-minimizing poisoning, we find that moving in $\ell_2$ normalized gradient directions is ineffective at reaching the required universal stop error [18], so we move in signed gradient directions instead (as is done for $\ell_\infty$ PGD attacks).

**Regions-4 and Regions-16 Noise.** Synthetic, random noises are also dataset and network independent. Thus, to demonstrate the strength of our method, we include three class-wise random noises in our

Table 1: **Dataset Independence.** AR noises are effective across a variety of datasets. For SVHN, STL-10, and CIFAR-10, we use the same 10 AR processes to generate sample-wise noises. We display clean test accuracy of RN-18 when trained using standard augmentations on different poisons. Class-wise poisons are marked with ○ and sample-wise poisons are marked with ●.

|  | SVHN | STL-10 | CIFAR-10 | CIFAR-100 |
|---|---|---|---|---|
| Clean | 96.40 | 83.65 | 93.62 | 75.30 |
| ● Error-Max [10] | 80.80 | 17.26 | 26.94 | 4.87 |
| ● Error-Min [18] | 96.82 | 80.71 | 16.84 | 74.58 |
| ○ Regions-4 | 9.80 | 41.36 | 20.75 | 9.14 |
| ○ Regions-16 | **6.39** | 31.21 | 15.75 | **2.99** |
| ○ Random Noise | 9.68 | 72.07 | 18.45 | 73.90 |
| ● Autoregressive (Ours) | 6.77 | **11.65** | **11.75** | 4.24 |

Table 2: **Architecture Independence.** CIFAR-10 test accuracy for a variety of model architectures trained on different poisons. Error-Max and Error-Min poisons are crafted using a RN-18.

|  | RN-18 | VGG-19 | GoogLeNet | MobileNet | EfficientNet | DenseNet | ViT |
|---|---|---|---|---|---|---|---|
| ● Error-Max [10] | 16.84 | 20.34 | 15.31 | 15.38 | 16.21 | 18.02 | 41.70 |
| ● Error-Min [18] | 26.94 | 22.39 | 32.18 | 21.36 | 23.86 | 28.21 | 40.95 |
| ○ Regions-4 | 20.75 | 25.39 | 25.25 | 22.23 | 18.64 | 24.21 | 35.18 |
| ○ Regions-16 | 15.75 | 19.86 | 12.97 | 19.67 | 16.94 | 24.16 | 25.49 |
| ○ Random Noise | 18.45 | 81.98 | 12.53 | 46.61 | 86.89 | **9.96** | 74.47 |
| ● AR (Ours) | **11.75** | **12.35** | **9.24** | **15.17** | **13.47** | 14.90 | **19.66** |

experiments. To generate what we a call a Regions-$p$ noise, we follow [39, 27]: we sample $p$ RGB vectors of size 3 from a Gaussian distribution and repeat each vector along height and width dimensions, resulting in a grid-like pattern of $p$ uniform cells or regions. Assuming a square image of side length $L$, a Regions-$p$ noise contains patches of size $\frac{L}{\sqrt{p}} \times \frac{L}{\sqrt{p}}$.

**Random Noise.** We also consider a class-wise random noise poison, where perturbations for each class are sampled from a Gaussian distribution.

## 4.2 AR Perturbations are Dataset and Architecture Independent

Unlike error-maximizing and error-minimizing poisons, AR poisons are not dataset-specific. One cannot simply take the perturbations from an error-maximizing or error-minimizing poison and apply the same perturbations to images of another dataset. Perturbations optimized using PGD are known to be relevant features, necessary for classification [10, 19]. Additionally, for both these methods, a crafting network trained on clean data is needed to produce reasonable gradient information. In contrast, AR perturbations are generated from dataset-independent AR processes. The same set of AR processes can be used to generate the same kinds of noise for images of new datasets. Building from this insight, one could potentially collect a large set of $K$ AR processes to perturb any dataset of $K$ or fewer classes, further showing the generality of our method.

In Table 1, we use the same 10 AR processes to generate noise for images of SVHN, STL-10, and CIFAR-10. AR poisons are, in all cases, either competitive or the most effective poison – a poison-trained RN-18 reaches nearly chance accuracy on STL-10 and CIFAR-10, and being the second-best on SVHN and CIFAR-100. The generality of AR perturbations to different kinds of datasets suggests that AR poisoning induces the most easily learned correlation between samples and their corresponding label.

We also evaluate the effectiveness of our AR poisons when different architectures are used for training. Recall that error-maximizing and error-minimizing poisoning use a crafting network to optimize the input perturbations. Because it may be possible that these noises are specific to the network architecture, we perform an evaluation of test set accuracy on CIFAR-10 after poison training VGG-19 [32], GoogLeNet [33], MobileNet [16], EfficientNet [34], DenseNet [17], and ViT [8]. Our ViT uses a patch size of 4. In Table 2, we show that Error-Max and Error-Min poisons generalize relatively

Table 3: **Strong Data Augmentations.** CIFAR-10 test accuracy of RN-18 when training using standard augmentations plus Cutout, CutMix, or Mixup on different poisons, where clean images are perturbed within an $\ell_2$ ball of size $\epsilon$.

|  |  | Standard Aug | +Cutout | +CutMix | +Mixup |
|---|---|---|---|---|---|
| • Error-Max [10] |  | 25.04 | 25.38 | 29.72 | 38.07 |
| • Error-Min [18] |  | 48.07 | 44.97 | 53.77 | 53.81 |
| ○ Regions-4 | $\epsilon = 0.5$ | 57.48 | 67.47 | 67.72 | 66.80 |
| ○ Regions-16 |  | 47.35 | 39.64 | 32.67 | 49.02 |
| ○ Random Noise |  | 93.76 | 93.66 | 91.80 | 94.16 |
| • Autoregressive (Ours) |  | **14.28** | **12.36** | **18.02** | **14.59** |
| • Error-Max [10] |  | 16.84 | 18.86 | 21.45 | 30.52 |
| • Error-Min [18] |  | 26.94 | 29.38 | 25.04 | 43.36 |
| ○ Regions-4 | $\epsilon = 1$ | 20.75 | 21.89 | 28.61 | 40.60 |
| ○ Regions-16 |  | 15.75 | 19.61 | 16.67 | 20.81 |
| ○ Random Noise |  | 18.45 | 12.61 | 12.63 | 23.64 |
| • Autoregressive (Ours) |  | **11.75** | **11.90** | **11.23** | **11.40** |

well across a range of related CNNs, but struggle with ViT, which is a transformer architecture. In contrast, our AR poison is effective across all CNN architectures and is the most effective poison against ViT. Our AR poison is much more effective over other poisons in almost all cases, achieving improvements over the next best poison of $4\%$ on RN-18, $5.8\%$ on ViT, and $7.5\%$ on GoogLeNet. The design of AR perturbations is meant to target the convolution operation, so it is surprising to see a transformer network be adversely affected. We believe our AR poison is particularly effective on GoogLeNet due to the presence of Inception modules that incorporate convolutions using various filter sizes. While our AR perturbations are generated using a $3 \times 3$ window, the use of various filter sizes may exaggerate their separability, as described in Section 3.3.

## 4.3 AR Perturbations Against Common Defenses

### 4.3.1 Data Augmentations and Smaller Perturbations

Our poisoning method relies on imperceptible AR perturbations, so it is conceivable that one could modify the data to prevent the learning of these perturbations. One way of modifying data is by using data augmentation strategies during training. In addition to standard augmentations like random crops and horizontal flips, we benchmark our AR poison against stronger augmentations like Cutout [7], CutMix [41], and Mixup [42] in Table 3. Generally, Mixup seems to be the most effective at disabling poisons. A RN-18 poison-trained using standard augmentations plus Mixup can achieve a boosts in test set performance of $13.68\%$ on Error-Max, $16.42\%$ on Error-Min, $19.85\%$ on Regions-4, $5.05\%$ on Regions-16, and $5.19\%$ on Random Noise. However, a RN-18 poison-trained on our AR poison ($\epsilon = 1$) using standard augmentations plus Cutout, CutMix, or Mixup cannot achieve *any* boost in test set performance.

We also present results for poisons using perturbations of size $\epsilon = 0.5$ to explore just how small perturbations can be made while still maintaining poisoning performance. Under standard augmentations, going from larger to smaller perturbations ($\epsilon = 1$ to $\epsilon = 0.5$), poison effectiveness drops by $8.2\%$ for Error-Max, $21.13\%$ for Error-Min, $36.73\%$ for Regions-4, and $31.6\%$ for Regions-16. Our AR poison achieves the smallest drop in effectiveness: only $2.53\%$. Random noise can no longer be considered a poison at $\epsilon = 0.5$ – it completely breaks for small perturbations. Under *all* strong data augmentation strategies at $\epsilon = 0.5$, AR poisoning dominates. For example, under Mixup, the best runner-up poison is Error-Max with an effectiveness that is more than $23\%$ lower than AR. Unlike all other poisons, AR poisoning is exceptionally effective for small perturbations.

Note that in all three augmentation strategies pixels are either dropped or scaled. Our method is unaffected by these augmentation strategies, unlike error-maximizing, error-minimizing, and other randomly noise poisons. Scaling an AR perturbation does not affect how the corresponding matching AR filter will respond,[2] and thus, the patterns remain highly separable regardless of perturbation size.

---

[2]See condition outlined in Lemma 3.1.

Table 4: **Adversarial Training.** CIFAR-10 test accuracy after adversarially training with different radii $\rho_a$. Top row shows performance of adversarial training on clean data. AR poisons remain effective for small $\rho_a$.

| | | $\rho_a$ | | |
|---|---|---|---|---|
| | 0.125 | 0.25 | 0.50 | 0.75 |
| Clean Data | 87.07 | 84.75 | 81.19 | 77.01 |
| • Error-Max [10] | $33.30_{\pm 0.14}$ | $72.27_{\pm 2.18}$ | $81.15_{\pm 3.58}$ | $78.73_{\pm 4.20}$ |
| • Error-Min [18] | $70.66_{\pm 0.41}$ | $84.80_{\pm 2.38}$ | $83.04_{\pm 3.24}$ | $79.11_{\pm 3.46}$ |
| ○ Regions-4 | $75.05_{\pm 0.35}$ | $81.23_{\pm 0.11}$ | $79.71_{\pm 0.05}$ | $76.47_{\pm 0.34}$ |
| ○ Regions-16 | $47.99_{\pm 0.25}$ | $71.43_{\pm 0.17}$ | $80.47_{\pm 0.10}$ | $76.65_{\pm 0.07}$ |
| ○ Random Noise | $86.31_{\pm 0.42}$ | $84.17_{\pm 0.20}$ | $\mathbf{80.11_{\pm 0.06}}$ | $\mathbf{76.26_{\pm 0.07}}$ |
| • Autoregressive (Ours) | $\mathbf{33.22_{\pm 0.77}}$ | $\mathbf{57.08_{\pm 0.75}}$ | $81.27_{\pm 2.61}$ | $79.07_{\pm 3.47}$ |

Table 5: **Mixing Poisons with Clean Data.** CIFAR-10 test accuracy when a proportion of clean data is used in addition to a poison. Top row shows test accuracy when training on only the clean proportion of the data; *i.e.* no poisoned data is used.

| | Clean Proportion | | | | |
|---|---|---|---|---|---|
| | 40% | 30% | 20% | 10% | 5% |
| Clean Only | 90.84 | 89.92 | 87.90 | 81.01 | 74.97 |
| • Error-Max [18] | $87.83_{\pm 0.74}$ | $86.83_{\pm 0.48}$ | $84.70_{\pm 0.61}$ | $81.63_{\pm 0.63}$ | $76.48_{\pm 1.72}$ |
| • Error-Min [10] | $88.32_{\pm 1.57}$ | $87.23_{\pm 0.84}$ | $84.56_{\pm 0.88}$ | $78.76_{\pm 1.83}$ | $67.82_{\pm 1.92}$ |
| ○ Regions-4 | $88.94_{\pm 0.85}$ | $86.75_{\pm 0.86}$ | $83.52_{\pm 0.20}$ | $78.23_{\pm 0.97}$ | $70.19_{\pm 3.16}$ |
| ○ Regions-16 | $88.03_{\pm 0.57}$ | $86.23_{\pm 0.68}$ | $\mathbf{83.01_{\pm 0.48}}$ | $76.52_{\pm 0.91}$ | $67.24_{\pm 1.72}$ |
| ○ Random Noise | $\mathbf{86.40_{\pm 1.24}}$ | $86.99_{\pm 0.19}$ | $84.98_{\pm 1.85}$ | $78.08_{\pm 0.94}$ | $70.69_{\pm 0.87}$ |
| • AR (Ours) | $87.63_{\pm 0.68}$ | $\mathbf{85.62_{\pm 0.62}}$ | $83.28_{\pm 0.90}$ | $\mathbf{76.13_{\pm 2.34}}$ | $62.69_{\pm 5.58}$ |

Additionally, AR filters contain values which sum to 0, so uniform regions of an image also produce a zero response.

### 4.3.2 Adversarial Training

Adversarial training has also been shown to be an effective counter strategy against $\ell_p$-norm constrained data poisons [18, 10, 9, 35]. Using adversarial training, a model trained on the poisoned data can achieve nearly the same performance as training on clean data [30]. However, adversarial training is computationally more expensive than standard training and leads to a decrease in test accuracy [21, 36] when the perturbation radius, $\rho_a$, of the adversary is large. In Table 4, we include adversarial training results on clean data to outline this trade-off where training at large $\rho_a$ comes at the cost of test accuracy. A recent line of work has therefore focused on developing better data poisoning methods that are robust against adversarial training [30, 37] at larger adversarial training radius $\rho_a$.

In Table 4, we compare performance of different poisons against adversarial training. We perform $\ell_2$ adversarial training with different perturbation radii, $\rho_a$, using a 7-step PGD attack with a step-size of $\rho_a/4$. We report error-bars by training three independent models for each run. We also show the performance of adversarial training on clean data. Data poisoning methods are fragile to adversarial training even when the training radius $\rho_a$ is smaller than poisoning radius $\epsilon$ [30, 37]. It is desirable for poisons to remain effective for larger $\rho_a$, because the trade-off between standard test accuracy and robust test accuracy would be exaggerated further. As shown in the Table 4, when the adversarial training radius $\rho_a$ increases, the poisons are gradually rendered ineffective. All poisons are nearly ineffective at $\rho_a = 0.5$. Our proposed AR perturbations remain more effective at smaller radius, *i.e.* $\rho_a = 0.125$ and $\rho_a = 0.25$ compared to all other poisons.

### 4.3.3 Mixing Poisons with Clean Data

Consider the scenario when not all the data can be poisoned. This setup is practical because, to a practitioner coming into control of poisoned data, additional clean data may be available through other sources. Therefore, it is common to evaluate poisoning performance using smaller proportions

of randomly selected poison training samples [10, 18, 30]. A poison can be considered effective if the addition of poisoned data hurts test accuracy compared to training on only the clean data. In Table 5, we evaluate the effectiveness of poisons using different proportions of clean and poisoned data. The top row of Table 5 shows test accuracy after training on only the subset of clean data, with no poisoned data present. We report error-bars by training four independent models for each run. Our AR poisons remain effective compared to other poisons even when clean data is mixed in. AR poisons are much more effective when a small portion of the data is clean. For example, when $5\%$ of data is clean, a model achieves ~$75\%$ accuracy when training on only the clean proportion, but using an additional $95\%$ of AR data leads to a ~$9\%$ decrease in test set generalization. Our results on clean data demonstrate that AR poisoned data is worse than useless for training a network, and a practitioner with access to the data would be better off not using it.

## 5 Conclusion

Using the intuition that simple noises are easily learned, we proposed the design of AR perturbations, noises that are so simple they can be perfectly classified by a 3-layer CNN where all parameters are manually-specified. We demonstrate that these AR perturbations are immediately useful and make effective poisons for the purpose of preventing a network from generalizing to the clean test distribution. Unlike other effective poisoning techniques that optimize error-maximizing or error-minimizing noises, AR poisoning does not need access to a broader dataset or surrogate network parameters. We are able to use the same set of 10 AR processes to generate imperceptible noises able to degrade the test performance of networks trained on three different 10 class datasets. Unlike randomly generated poisons, AR poisons are more potent when training using a new network architecture or strong data augmentations like Cutout, CutMix, and Mixup. Against defenses like adversarial training, AR poisoning is competitive or among the best for a range of attack radii. Finally, we demonstrated that AR poisoned data is worse than useless when it is mixed with clean data, reducing likelihood that a practitioner would want to include AR poisoned data in their training dataset.

## Acknowledgments and Disclosure of Funding

This material is based upon work supported by the National Science Foundation under Grant No. IIS-1910132 and Grant No. IIS-2212182, and by DARPA's Guaranteeing AI Robustness Against Deception (GARD) program under #HR00112020007. Pedro is supported by an Amazon Lab126 Diversity in Robotics and AI Fellowship. Any opinions, findings, and conclusions or recommendations expressed in this material are those of the author(s) and do not necessarily reflect the views of the National Science Foundation.

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
