# OpenReview forum: "Autoregressive Perturbations for Data Poisoning"
_NeurIPS.cc/2022/Conference — NeurIPS 2022 Accept_

### Official Review · Reviewer_wQLZ · 2022-07-06

**Rating:** 5
**Confidence:** 4
**Soundness:** 2 fair
**Presentation:** 3 good
**Contribution:** 2 fair

**Summary:**

This paper proposes to use autoregressive processes to generate perturbations for data poisoning. The generated perturbations, despite looking complex, are actually very simple. One advantage of the proposed method is that its generated perturbations are dataset and architecture independent. The paper evaluates the proposed method on multiple datasets and networks, showing the effectiveness of the perturbations when the poison rate is high.

**Questions:**

1. Is the proposed method only applicable to $\ell_2$ norm?
2. The proposed attack requires high poison rate to be effective?
3. Is the proposed method only applicable to computer vision tasks?

**Limitations:**

1. The paper only studies $\ell_2$ norm.
2. The poison rate is high. The lowest poison rate studied in the paper is 0.6.
3. The theoretical analysis is not sufficient. The relation between Lemma 3.1 and the effectiveness of the proposed method in poisoning attacks is not obvious.


**Strengths And Weaknesses:**

The strengths of this paper include
1. The proposed attack method is interesting.
2. The proposed method has good transferability.

But I still have the following concerns:

1. Is the proposed method only applicable to $\ell_2$ norm? The paper uses the sentence "We measure AR perturbations
in $\ell_2$ because measuring in $\ell_\infty$ would underestimate the extent to which these perturbations are less perceptible than purely  $\ell_\infty$ random noise" to explain why it uses $\ell_2$ norm. But this sentence is hard to follow, and this short explanation is not convincing. The paper should provide more clear and convincing explanation about why it *only* uses $\ell_2$ norm.

2. The proposed attack requires high poison rate to be effective? In most experiments, the paper uses poison rate 1. In Table, the lowest poison rate is 0.6. The assumption of high poison rate is very strong. In practice, if the data is collected from multiple sources, then the attack is not effective? In the case that the data is collected from one source (the adversary), the entity who trains the model would be more cautious about the quality of data due to the high risk when the data only comes from one source.

3. Section 3.3 is not easy to follow, and the logic is not very clear. I think Section 3.3 is one of the most important parts in the paper since it explains why the proposed method works. After reading Section 3.3, I am still very confused. The relation between Lemma 3.1 and the effectiveness of the proposed method in *poisoning attacks* is not obvious.

---

> ### Author Response · Authors · 2022-08-02
> **Response to Reviewer wQLZ [Part 1]**
>
> We would like to thank the reviewer for their thoughtful feedback, and for considering our method “interesting.”
> ### Responses to concerns
> > Is the proposed method only applicable to $\ell_2$ norm?
>
> We initially measured the $\ell_2$-norm because our poisons are not optimized for a specific $\ell_p$ constraint. AR perturbations may have single entries which are high and violate a strict $\ell_{\infty}$ constraint. Still, our proposed perturbations can be projected onto any $\ell_p$-norm ball, including $\ell_{\infty}$. To demonstrate that AR poisoning *can* work in the $\ell_{\infty}$-norm constrained setting, we provide CIFAR-10 test accuracy for a RN-18, where perturbations are of size $\frac{8}{255}$ in $\ell_{\infty}$-norm.
>
> |                       | Standard Aug | +Cutout | +CutMix | +Mixup |
> |-----------------------|--------------|---------|---------|--------|
> | AR (Ours)              |     20.49    |  26.93  |  17.08  |  15.22 |
>
> Importantly, how one fits an AR perturbation $\delta$ within the constraint that $\lVert \delta \rVert_{\infty} \leq \epsilon = \frac{8}{255}$ affects performance. Here, we simply compute $\epsilon \frac{\delta}{\lVert \delta \rVert_{\infty}}$, which may be suboptimal, but we haven’t had the chance to explore different options. Clipping values or taking the scaled sign of $\delta$ would make the perturbation no longer autoregressive. We agree that the addition of this experiment makes our work more complete, so we have included this new table in the Appendix Table 6.
>
> > The proposed attack requires a high poison rate to be effective?
>
> The goal of our work is to prevent others from using our poisoned data to increase the performance of their models, or to train their models in the first place, as in [4, 5, 6, 7] which also poison a very high fraction or all of the data.  We have updated Table 4 (with standard deviation over 4 independent runs), and we see that, in all but one case, adding our poisoned data to clean data reduces the test accuracy of models, so we can indeed effectively prevent others from leveraging our data.  This goal, namely preventing the addition of poisoned data from boosting accuracy, is standard in the literature [4, 5, 6, 7].
>
>
> | Poison/Clean Proportion | 40%            | 30%            | 20%            | 10%            | 5%           |
> |-------------------|----------------|----------------|----------------|----------------|----------------|
> | Clean             | 90.29 &pm; 0.38 | 88.57 &pm; 0.34 | 85.17 &pm; 1.10 | 74.65 &pm; 4.41 | 70.20 &pm; 5.22 |
> | AR (Ours)         | 87.63 &pm; 0.68 | 85.62 &pm; 0.62 | 83.28 &pm; 0.90 | 76.13 &pm; 2.34 | 62.69 &pm; 5.58 |
> | Difference         | -2.66               | -2.95               | -1.89                | +1.48              | -7.51             |
>
> > In Table 4, the lowest poison rate is 0.6
>
> Taking the reviewer’s suggestion, we have also performed new experiments for when the poison proportion is under 60% (i.e. when clean proportion exceeds 40%). We find that it is better to train a network without AR poisoned data when the clean proportion is 50%, 60%, and even 70%, as desired. The goal is to render the data useless for generalization, while maintaining the content of the image.
>
> |  Poison/Clean Proportion          | 90%            | 80%            | 70%            | 60%            | 50%            |
> |------------|----------------|----------------|----------------|----------------|----------------|
> | Clean        | 91.89 &pm; 0.51 | 91.77 &pm; 0.15 | 91.18 &pm; 0.16 | 91.10 &pm; 0.32 | 90.86 &pm; 0.28 |
> | AR (Ours)  | 92.37 &pm; 0.16 | 91.79 &pm; 0.14 | 91.05 &pm; 0.32 | 90.46 &pm; 0.32 | 89.28 &pm; 0.52 |
> | Random     | 92.68 &pm; 0.39 | 92.08 &pm; 0.42 | 91.94 &pm; 0.22 | 90.58 &pm; 1.22 | 89.78 &pm; 0.57 |
> | Regions-4  | 92.43 &pm; 0.26 | 92.15 &pm; 0.17 | 91.47 &pm; 0.15 | 91.32 &pm; 0.77 | 90.16 &pm; 1.14 |
> | Regions-16 | 92.04 &pm; 0.27 | 91.76 &pm; 0.39 | 91.46 &pm; 0.22 | 90.08 &pm; 1.03 | 89.75 &pm; 0.74 |
> | Error-Max  | 91.26 &pm; 0.23 | 91.18 &pm; 0.47 | 90.68 &pm; 0.83 | 90.12 &pm; 0.50 | 88.76 &pm; 0.70 |
> | Error-Min  | 91.99 &pm; 0.16 | 91.71 &pm; 0.72 | 91.98 &pm; 0.17 | 91.28 &pm; 0.89 | 89.83 &pm; 0.48 |
>
> [4] Unlearnable Examples: Making Personal Data Unexploitable, ICLR 2021 \
> [5] Adversarial Examples Make Strong Poisons, NeurIPS 2021 \
> [6] Learning to Confuse: Generating Training Time Adversarial Data with Auto-Encoder, NeurIPS 2019 \
> [7] TensorClog: An Imperceptible Poisoning Attack on Deep Neural Network Applications, IEEE Access 2019

---

> > ### Author Response · Authors · 2022-08-02
> > **Response to Reviewer wQLZ [Part 2]**
> >
> > > Section 3.3 is not easy to follow…The theoretical analysis is not sufficient. The relation between Lemma 3.1 and the effectiveness of the proposed method in poisoning attacks is not obvious.
> >
> > We thank the reviewer for pointing this out. We have rewritten Section 3.3 for clarity, focusing on our logic and its relationship to Lemma 3.1. Here is the relevant portion of the new text: A signal that is easily interpolated by a network will be quickly identified and used as a “shortcut,” whereas complex and unpredictable patterns may not be learned until after a network has already extracted useful content-based features [8]. We propose a simple hypothesis: if there exists a simple CNN that can classify autoregressive signals perfectly, then these signals will be easy to learn. By showing that AR filters exist, Lemma 3.1 helps us define the simple CNN that classifies AR signals perfectly. Our experiments demonstrate that our method, motivated by our simple hypothesis, is effective.
> >
> > ### Responses to questions
> > > Is the proposed method only applicable to computer vision tasks?
> >
> > We only develop perturbations for images, but an AR perturbation can be crafted for any continuous signal. We speculate that our method could work for audio classification as well. We show that AR patterns are easily learned by CNNs, and they are applicable to any setting where you would use a CNN.
> >
> > Thank you again for your feedback. We think that your suggestions have improved our paper. We made a significant effort to address your questions, and would appreciate it if you would consider raising your score in light of our response. Please let us know if you have any additional questions we can address.
> >
> > [8] The pitfalls of simplicity bias in neural networks, NeurIPS 2020

---

> ### Author Response · Authors · 2022-08-08
> **Following Up with Reviewer wQLZ**
>
> Thank you again for your thoughtful review. Does our response help address your feedback? We would appreciate the opportunity to engage further if needed.

---

> > ### Comment · Reviewer_wQLZ · 2022-08-09
> > **Thank you for your comprehensive response**
> >
> > The response addressed most of my concerns. So I raised my rating.
> > One remaining concern is that, although a simple signal is easily interpolated by a network, it may also be easily eliminated by well-designed denoising techniques, e.g., a denoising method that is specifically designed for the AR noise. Although difficult patterns are hard to learn, they may be more robust against denoising methods.

---

> > > ### Author Response · Authors · 2022-08-10
> > > **Response to remaining concern**
> > >
> > > > although a simple signal is easily interpolated by a network, it may also be easily eliminated by well-designed denoising techniques
> > >
> > > Thank you for your additional feedback.  Designing denoisers for autoregressive perturbations, which are generated using AR processes unknown to the victim, requires that the denoiser be agnostic to the exact AR process. Even if AR coefficients were leaked, there would still be 372 floating point values unknown to the victim (because we sample our starting signal from a Gaussian for a 32x32x3 image and an AR process that uses a window size 3x3) (Figure 3, Left). Recovering or removing perturbations is a challenging direction for future work, but developing novel defense techniques is beyond the scope of this paper.
> > >
> > > Furthermore, while adding AR perturbations to training data makes the data easy to fit, estimating the noise from poisoned data may be very challenging, perhaps no less challenging than estimating perturbations added under other indiscriminate poisoning attacks, such as error-max perturbations [1].  We emphasize that how easy the noised training data is to fit is **not** related to the difficulty of recovering the clean data after the noise is applied. Error-max and Error-min perturbations are “almost linearly separable” [2] and yet denoisers which remove perturbations under indiscriminate poisoning attacks to recover model performance remain elusive. The possibility of denoisers as a defense can just as easily be raised against other indiscriminate poisoning methods. We do think this is an interesting direction, and defenses are worth pursuing. If we have addressed your feedback, we hope you will consider increasing your score.
> > >
> > > [1] Adversarial Examples Make Strong Poisons, NeurIPS 2021
> > >
> > > [2] Availability Attacks Create Shortcuts, KDD 2022

---

> > > > ### Comment · Reviewer_wQLZ · 2022-08-10
> > > > **Thank you for the response**
> > > >
> > > > Thank you for the response. I already increased the score to 5 (borderline accept).
> > > >
> > > > What the paper proposes is a defense to make the data unlearnable, so developing novel defenses means developing adaptive attacks? In some related literature, designing adaptive attacks is considered necessary to verify the effectiveness of defenses, and lacking adaptive attacks is a weakness or limitation mentioned in many reviews. So the limitation remains. But I do not think it is a big limitation. So I raise the score to 5 since the contributions overweigh the weakness and limitation.

---

### Official Review · Reviewer_GeFk · 2022-07-09

**Rating:** 7
**Confidence:** 5
**Soundness:** 3 good
**Presentation:** 3 good
**Contribution:** 3 good

**Summary:**

This paper proposed autoregressive poisoning techniques to protect data from being exploited by unauthorized machine learning models. The proposed method does not rely on optimizations while generic towards different model architectures and different datasets. This paper also provides insight into why the proposed method is effective.


**Questions:**

- In experiments section line210:  "We say that poisoning effectiveness drops from setup A to setup B if the network from poison-trained on setup B has higher test set accuracy than the network poison-trained on setup A. " I find this is confusing.
- For experiments in Table 4, for clean only, is it the same subset of data as in mixing poisons/clean?

**Limitations:**

Please address the potential limitations in the Strengths And Weaknesses section.

**Strengths And Weaknesses:**

Strengths:
- The proposed method is efficient and technically sound. Existing works rely on optimizations which is the bottleneck. The proposed method does not rely on optimizations, and the parameters for AR are easy to find.
- The existing works are also shown that do not transfer well between model architectures or datasets. Experiment results show that one set of AR is generic across different architectures or datasets.
- The efficient and generic can be very practical considering real-world applications.
- Experiment results also demonstrated AR generated unlearnable examples are more robust towards augmentations.

---
Limitations:
- Once the data is released, the defender may not modifies the data anymore, and the model trainer can retroactively apply new models/methods [1]. The adaptive case should be carefully examined. Consider that if the parameters for AR are leaked, can it be used to recover the original image? Or if a portion of the clean images are leaked, using pair of clean and unlearnable versions, is it possible to reverse the AR process?
- In section 3.3, the assertion that the noises are easy to learn is more effective for poisoning, this could also mean they are easy to detect. Such as calculating sample-specific loss at the end of each training epoch. Although only detecting such samples does not make them "learnable," but adaptive method (if there are any) can be applied to these samples. Or the model trainer could wait for future advancement for the recovery method as mentioned in [1].

[1] Data Poisoning Won’t Save You From Facial Recognition, ICML 2021 Workshop AML

---
After the author's response, I increased my rating score to 7. My main concerns over possible reverse operation if parameters are leaked have been well addressed.

---

> ### Author Response · Authors · 2022-08-02
> **Response to Reviewer GeFk**
>
> We thank the reviewer for their time and for indicating that our method is “efficient,” “technically sound,” and “very practical considering real-world applications.” Below, we respond to concerns and then answer posed questions.
>
> ### Responses to weaknesses
>
> > Consider that if the parameters for AR are leaked, can it be used to recover the original image?
>
> No, the leaked parameters are not sufficient to recover the original image. Consider that we start from a random Gaussian starting signal (Figure 2.1) for every image, which is independent from the AR process and not shared, before applying an AR process. This means that even if AR coefficients were leaked, there would still be 372 floating point values unknown to the victim (for a 32x32x3 image with an AR process that uses a window size 3x3). While AR perturbations from the same AR process may look similar, they are unique (Figure 3, Left).
>
> > but adaptive method (if there are any) can be applied to these samples. Or the model trainer could wait for future advancement for the recovery method
>
> While there could be potential detection techniques, as the reviewer suggested, developed for different data poisons, removing these AR poisons is not trivial. We agree this would be a topic for future work.
>
> ### Responses to questions
> > In experiments section line210: "We say that poisoning effectiveness drops from setup A to setup B if the network from poison-trained on setup B has higher test set accuracy than the network poison-trained on setup A. " I find this is confusing.
>
> This sentence was mainly used to describe “poisoning effectiveness.” We agree this sentence was confusing, so we have removed it. Other sentences already define what is “effective” and how to read numbers in the experimental section.
>
> > For experiments in Table 4, for clean only, is it the same subset of data as in mixing poisons/clean?
>
> The selected clean subset in Table 4 is different, and i.i.d sampled for clean-only training and for each poison. However, we do not believe this impacts the trends we observe. Below, we provide an updated table, where we report results over 4 independent runs, and observe the same trends.
> | Poison/Clean Proportion | 40%            | 30%            | 20%            | 10%            | 5%           |
> |-------------------|----------------|----------------|----------------|----------------|----------------|
> | Clean             | 90.29 &pm; 0.38 | 88.57 &pm; 0.34 | 85.17 &pm; 1.10 | 74.65 &pm; 4.41 | 70.20 &pm; 5.22 |
> | AR (Ours)         | 87.63 &pm; 0.68 | 85.62 &pm; 0.62 | 83.28 &pm; 0.90 | 76.13 &pm; 2.34 | 62.69 &pm; 5.58 |
> | Random            | 86.40 &pm; 1.24 | 86.99 &pm; 0.19 | 84.98 &pm; 1.85 | 78.08 &pm; 0.94 | 70.69 &pm; 0.87 |
> | Regions-4         | 88.94 &pm; 0.85 | 86.75 &pm; 0.86 | 83.52 &pm; 0.20 | 78.23 &pm; 0.97 | 70.19 &pm; 3.16 |
> | Regions-16        | 88.03 &pm; 0.57 | 86.23 &pm; 0.68 | 83.01 &pm; 0.48 | 76.52 &pm; 0.91 | 67.24 &pm; 1.72 |
> | Error-Max         | 87.83 &pm; 0.74 | 86.83 &pm; 0.48 | 84.70 &pm; 0.61 | 81.63 &pm; 0.63 | 76.48 &pm; 1.72 |
> | Error-Min         | 88.32 &pm; 1.57 | 87.23 &pm; 0.84 | 84.56 &pm; 0.88 | 78.76 &pm; 1.83 | 67.82 &pm; 1.92 |
>
>
> When mixing in clean data we observe our method always leads to a decrease in test accuracy when poisoned data is added. Put another way, in all but one case, it is better to exclude AR poisoned data than to use it for training. AR poisoning also performs better than all the other poisons when a small amount of clean data (5% or 10% clean data) is mixed in.

---

> > ### Comment · Reviewer_GeFk · 2022-08-05
> > **Thanks for addressing my concerns.**
> >
> > Thanks for the detailed clarification. All concerns have been addressed.

---

### Official Review · Reviewer_FXVx · 2022-07-11

**Rating:** 6
**Confidence:** 4
**Soundness:** 3 good
**Presentation:** 3 good
**Contribution:** 3 good

**Summary:**

This paper proposes a new data poisoning attack to prevent data scraping. The proposed method adds class conditional autoregressive (AR) noise to training data to prevent people from using the data for training, and the method is data and model independent, which means that the same noise can be used to poison different datasets and models of different architectures.

The intuition behind the idea is that easy to learn noise is more effective at data poisoning, and AR noise generated in the proposed way is easy for neural network to learn. The authors show that a manually specified 3-layer CNN with AR filter can easily learn class information from the AR noise. Experiments on four benchmark datasets (CIFAR10, STL10, SVHN, CIFAR100) show that the proposed method performs better than other four baselines (Error-min, Error-max, Regions, Random noise).

**Questions:**

About the process of AR noise generation:

- It is clear how to generate AR noise at the beginning inside the sliding window. How about the subsequent steps? Take the example in Figure 2 as an example, if the sliding window slides one step to the right, there are three values to be generated. Are $x_{t-7}$ up to $x_t$ used to generate the next one ($x_{t+1}$)? Then $x_{t-6}$ up to $x_{t+1}$ are used to generate $x_{t+2}$, and so on.

**Limitations:**

The author points out that the method does not perform well against adversarial training and experiments show that when evaluated using a mix of poisoned and clean data, the performance is also not good.

**Strengths And Weaknesses:**

Strengths:

- The proposed method is novel as autoregressive process hasn't been used before to do data poisoning. The method is easy to implement and the same AR coefficients can be used for different datasets and architectures as long as the numbers of classes are the same. Though code is not available, pseudo code (algorithms) and implementation details are provided. It is better that if actual code can be provided for reproduction of the results.

- The paper is well-written and easy to follow. Empirical results on four different datasets show that the method performs better than other baselines, both under normal setting and defense settings.

Weakness:

- Though the proposed method performs better than other baselines compared in the paper, when tested against adversarial training, the performance is not satisfactory. It performs similarly to other baselines under this setting and the poisoning effect is not good, especially when the radii is large.

- As pointed out in the paper, assuming that all the data can be poisoned is not realistic. In section 4.3.3, the poisoning methods are evaluated using a mix of poisoned and clean data. Under this setting, the performance of the proposed method is not good and similar to those of other baselines.

---

> ### Author Response · Authors · 2022-08-02
> **Response to Reviewer FXVx [Part 1]**
>
> We thank the reviewer for their thorough evaluation and for mentioning that our paper is “novel…well-written and easy to follow.”
> ### Responses to weaknesses
> > When tested against adversarial training, the performance is not satisfactory
>
> We have updated Table 5 (Adversarial Training) by performing additional runs, and we observe that our proposed method is statistically more effective for perturbation radius $\rho_a = 0.125$ and $\rho_a=0.25$. For every cell in the table, we report mean CIFAR-10 test accuracy over 2 additional models (for a total of 3 models).
>
> When considering larger perturbation radii for adversarial training, it is important to recall that adversarial training monotonically decreases test accuracy as the perturbation radius, $\rho_a$ becomes large [1, 2]. Additionally, recent work has shown that adversarial training is an effective defense for several data poisons [3]. Still, our method can better defend against adversarial training at small radii, and is competitive in the case when the radius is large. Our updated Table 5 is below:
>
> | Attack \\ $\rho_a$  | 0.125           | 0.25            | 0.5             | 0.75            |
> |:----------:|:---------------:|:---------------:|:---------------:|:---------------:|
> | AR (Ours)  | 33.22 &pm; 0.77 | 57.08 &pm; 0.75 | 81.27 &pm; 2.61 | 79.07 &pm; 3.47 |
> | Random     | 86.31 &pm; 0.42 | 84.17 &pm; 0.20 | 80.11 &pm; 0.06 | 76.26 &pm; 0.07 |
> | Regions-4  | 75.05 &pm; 0.35 | 81.23 &pm; 0.11 | 79.71 &pm; 0.05 | 76.47 &pm; 0.34 |
> | Regions-16 | 47.99 &pm; 0.25 | 71.43 &pm; 0.17 | 80.47 &pm; 0.10 | 76.65 &pm; 0.07 |
> | Error-Max  | 33.30 &pm; 0.14 | 72.27 &pm; 2.18 | 81.15 &pm; 3.58 | 78.73 &pm; 4.20 |
> | Error-Min  | 70.66 &pm; 0.41 | 84.80 &pm; 2.38 | 83.04 &pm; 3.24 | 79.11 &pm; 3.46 |
>
> > Assuming that all the data can be poisoned is not realistic
>
> The goal of our work is to prevent others from using our poisoned data to increase the performance of their models, or to train their models in the first place, as in [4, 5, 6, 7] which also poison a very high fraction or all of the data.  In Table 4 we see that, in all but one case, adding our poisoned data to clean data reduces the test accuracy of models, so that we can indeed effectively prevent others from leveraging our data.  This goal, namely preventing the addition of poisoned data from boosting accuracy, is standard in the literature [4, 5, 6, 7].
>
> | Poison/Clean Proportion | 40%            | 30%            | 20%            | 10%            | 5%           |
> |-------------------|----------------|----------------|----------------|----------------|----------------|
> | Clean             | 90.29 &pm; 0.38 | 88.57 &pm; 0.34 | 85.17 &pm; 1.10 | 74.65 &pm; 4.41 | 70.20 &pm; 5.22 |
> | AR (Ours)         | 87.63 &pm; 0.68 | 85.62 &pm; 0.62 | 83.28 &pm; 0.90 | 76.13 &pm; 2.34 | 62.69 &pm; 5.58 |
> | Difference         | -2.66               | -2.95               | -1.89                | +1.48              | -7.51             |
>
> Taking the reviewer’s suggestion, we have also performed new experiments when the poison proportion is under 60% (i.e. when clean proportion exceeds 40%). We find that it is better to train a network without AR poisoned data when the clean proportion is 50%, 60%, and even 70%, as desired. The goal is to render the data useless for generalization, while maintaining the content of the image.
>
> |  Poison/Clean Proportion          | 90%            | 80%            | 70%            | 60%            | 50%            |
> |------------|----------------|----------------|----------------|----------------|----------------|
> | Clean        | 91.89 &pm; 0.51 | 91.77 &pm; 0.15 | 91.18 &pm; 0.16 | 91.10 &pm; 0.32 | 90.86 &pm; 0.28 |
> | AR (Ours)  | 92.37 &pm; 0.16 | 91.79 &pm; 0.14 | 91.05 &pm; 0.32 | 90.46 &pm; 0.32 | 89.28 &pm; 0.52 |
> | Random     | 92.68 &pm; 0.39 | 92.08 &pm; 0.42 | 91.94 &pm; 0.22 | 90.58 &pm; 1.22 | 89.78 &pm; 0.57 |
> | Regions-4  | 92.43 &pm; 0.26 | 92.15 &pm; 0.17 | 91.47 &pm; 0.15 | 91.32 &pm; 0.77 | 90.16 &pm; 1.14 |
> | Regions-16 | 92.04 &pm; 0.27 | 91.76 &pm; 0.39 | 91.46 &pm; 0.22 | 90.08 &pm; 1.03 | 89.75 &pm; 0.74 |
> | Error-Max  | 91.26 &pm; 0.23 | 91.18 &pm; 0.47 | 90.68 &pm; 0.83 | 90.12 &pm; 0.50 | 88.76 &pm; 0.70 |
> | Error-Min  | 91.99 &pm; 0.16 | 91.71 &pm; 0.72 | 91.98 &pm; 0.17 | 91.28 &pm; 0.89 | 89.83 &pm; 0.48 |
>
> [1] Theoretically Principled Trade-off between Robustness and Accuracy, ICML 2019 \
> [2] Robustness May Be at Odds with Accuracy, ICLR 2019 \
> [3] Better Safe Than Sorry: Preventing Delusive Adversaries with Adversarial Training, NeurIPS 2021 \
> [4] Unlearnable Examples: Making Personal Data Unexploitable, ICLR 2021 \
> [5] Adversarial Examples Make Strong Poisons, NeurIPS 2021 \
> [6] Learning to Confuse: Generating Training Time Adversarial Data with Auto-Encoder, NeurIPS 2019 \
> [7] TensorClog: An Imperceptible Poisoning Attack on Deep Neural Network Applications, IEEE Access 2019 \

---

> > ### Author Response · Authors · 2022-08-02
> > **Response to Reviewer FXVx [Part 2]**
> >
> > > It is better that if actual code can be provided for reproduction of the results
> >
> > We have uploaded our code repository as supplementary material to our submission. It contains documentation and example Jupyter notebooks.
> >
> > ### Responses to questions
> > > It is clear how to generate AR noise at the beginning inside the sliding window. How about the subsequent steps?
> >
> > Taking Figure 2.2 as an example, if the sliding window slides one step to the right, there is actually only one value (the next white grid cell) to be computed, $x_t$. Equation 5 is applied independently within every window. Put differently, for every window, the value $x_{t-8}$ is always at the top left corner of the window, the value $x_{t-6}$ is always the top right corner, etc. and $x_{t}$ is always the bottom right corner.

---

### Official Review · Reviewer_Jf4X · 2022-07-11

**Rating:** 6
**Confidence:** 4
**Soundness:** 3 good
**Presentation:** 3 good
**Contribution:** 3 good

**Summary:**

Paper considers a setting of `unlearnable examples' where a given dataset is perturbed in a way to make it hard to learn the true task. In essence, data here gets perturbed with correlated noise such that when learning is attempted, models learn to focus on the noise rather than on the true features useful for generalisation. While prior work focused on generating class-wise poisons, in this work the noise is generated per sample using a Markov process, producing linearly separatable noise. Paper thoroughly evaluates the setting and demonstrates that the approach effectively stops generalisation when the whole dataset is poisoned and struggles in a similar way in presence of adversrail training or dilution with clean data.

**Questions:**

Thank you very much for the paper, it is a very interesting read! I only have a handful of questions:

1. Are tables 4 and 5 computed over a number of models? Given how close the numbers are, it would be great to know if the differences are observed on distributional level, not just per model
2. Given that we can produce arbitrary correlated noise of different flavors, how should one think about it? What is the fundamental difference between the noises in the related literature and the one produced in the paper? This naturally leads to my final question.
3. Given the argument of easier learnability of different noises, it turns the question to how much observed behaviour is an artifact of the optimisation procedure itself? Did you try running the experiments with different lr/optimiser options?

Minor:
* Punctuation missing around eqs in some places

**Strengths And Weaknesses:**


Strengths:
+ Interesting setting
+ Idea of hardness of learning is rather fascinating

Weaknesses:
+ Unclear how much of the evaluation is an artifact of chosen optimisation hyperparameters
+ Unclear how one compares performance of different correlated noises

---

> ### Author Response · Authors · 2022-08-02
> **Response to Reviewer Jf4X [Part 1]**
>
> We thank the reviewer for their feedback, for asking questions which will improve our work, and for referring to our method as “fascinating.” Below, we respond to concerns and then answer posed questions.
>
> ### Responses to weaknesses
> > Unclear how much of the evaluation is an artifact of chosen optimisation hyperparameters
>
> To test the effect of optimizer and learning rate, we have run additional experiments to confirm the effectiveness of our method. Specifically, we consider 3 optimizers: SGD, SGD+Momentum ($\beta=0.9$ and Adam ($\beta_1=0.9$, $\beta_2=0.999$). We also consider 3 different learning-rate schedules: cosine learning-rate, single-step decay (epoch 50) and multi-step decay (epoch 50, epoch 75) with decay factor of 0.1. Thus, there are a total of 9 optimizer and learning rate combinations. In the following table, for each poison, we report mean CIFAR-10 test accuracy over the 9 combinations of optimizers and learning rates. Our proposed method remains nearly unaffected by choice of hyperparameters, and performs better than all other poisons.
>
> | AR (Ours)      | Random          | Regions-4     | Regions-16     | Error-Max    | Error-Min       |
> |----------------|-----------------|---------------|----------------|--------------|-----------------|
> | 12.23 &pm; 1.22 | 33.39 &pm; 31.24 | 22.89 &pm; 7.6 | 18.73 &pm; 5.52 | 16.6 &pm; 2.8 | 22.96 &pm;  7.16 |
>
> > Unclear how one compares performance of different correlated noises
>
> All the poisons we consider use the ground truth label as a dependent variable to generate the perturbations. Thus, poisons we consider contain perturbations that may be correlated with the ground truth. While there is an intractable number of possible noise patterns that could be used to perturb data and induce a correlation, we prove that AR perturbations have a particular structure which allows a manually-specified CNN filter to detect it perfectly (Lemma 3.1). Thus, AR perturbations are correlated noise designed for convolutional layers. By making sure that poisons we consider are bounded by the same $\ell_2$ perceptibility constraint, we are able to compare performance between different kinds of correlated noise.

---

> > ### Author Response · Authors · 2022-08-02
> > **Response to Reviewer Jf4X [Part 2]**
> >
> > ### Responses to questions
> > > Are tables 4 and 5 computed over a number of models?
> >
> > We have run the results of Table 4 and 5 over a number of models as suggested by the reviewer. We report these below.
> >
> > For Table 5 (Adversarial Training), we observe that our proposed method is statistically more effective for perturbation radius $\rho_a = 0.125$ and $\rho_a = 0.25$. For every cell in the table, we report mean CIFAR-10 test accuracy over 2 additional models (for a total of 3 models). When considering larger perturbation radii for adversarial training, it is important to recall that adversarial training monotonically (and steeply) decreases test accuracy as the perturbation radius, $\rho_a$ becomes large [1,2]. Additionally, recent work has shown that adversarial training is an effective defense for several data poisons [3]. Still, our method can better defend against adversarial training at small radii, and is competitive in the case when the radius is large. The results are shown below:
> >
> > | Attack \\ $\rho_a$  | 0.125           | 0.25            | 0.5             | 0.75            |
> > |:----------:|:---------------:|:---------------:|:---------------:|:---------------:|
> > | AR (Ours)  | 33.22 &pm; 0.77 | 57.08 &pm; 0.75 | 81.27 &pm; 2.61 | 79.07 &pm; 3.47 |
> > | Random     | 86.31 &pm; 0.42 | 84.17 &pm; 0.20 | 80.11 &pm; 0.06 | 76.26 &pm; 0.07 |
> > | Regions-4  | 75.05 &pm; 0.35 | 81.23 &pm; 0.11 | 79.71 &pm; 0.05 | 76.47 &pm; 0.34 |
> > | Regions-16 | 47.99 &pm; 0.25 | 71.43 &pm; 0.17 | 80.47 &pm; 0.10 | 76.65 &pm; 0.07 |
> > | Error-Max  | 33.30 &pm; 0.14 | 72.27 &pm; 2.18 | 81.15 &pm; 3.58 | 78.73 &pm; 4.20 |
> > | Error-Min  | 70.66 &pm; 0.41 | 84.80 &pm; 2.38 | 83.04 &pm; 3.24 | 79.11 &pm; 3.46 |
> >
> >
> > For Table 4 (Mixing Poisons with Clean Data), we run 3 additional models for a total of 4 models. When mixing in clean data we observe our method always leads to a decrease in test accuracy when poisoned data is added. Put another way, in all but one case, it is better to exclude AR poisoned data than to use it for training. AR poisoning also performs better than all the other poisons when a small amount of clean data (5% or 10% clean data) is mixed in.
> >
> > | Poison/Clean Proportion | 40%            | 30%            | 20%            | 10%            | 5%           |
> > |-------------------|----------------|----------------|----------------|----------------|----------------|
> > | Clean             | 90.29 &pm; 0.38 | 88.57 &pm; 0.34 | 85.17 &pm; 1.10 | 74.65 &pm; 4.41 | 70.20 &pm; 5.22 |
> > | AR (Ours)         | 87.63 &pm; 0.68 | 85.62 &pm; 0.62 | 83.28 &pm; 0.90 | 76.13 &pm; 2.34 | 62.69 &pm; 5.58 |
> > | Random            | 86.40 &pm; 1.24 | 86.99 &pm; 0.19 | 84.98 &pm; 1.85 | 78.08 &pm; 0.94 | 70.69 &pm; 0.87 |
> > | Regions-4         | 88.94 &pm; 0.85 | 86.75 &pm; 0.86 | 83.52 &pm; 0.20 | 78.23 &pm; 0.97 | 70.19 &pm; 3.16 |
> > | Regions-16        | 88.03 &pm; 0.57 | 86.23 &pm; 0.68 | 83.01 &pm; 0.48 | 76.52 &pm; 0.91 | 67.24 &pm; 1.72 |
> > | Error-Max         | 87.83 &pm; 0.74 | 86.83 &pm; 0.48 | 84.70 &pm; 0.61 | 81.63 &pm; 0.63 | 76.48 &pm; 1.72 |
> > | Error-Min         | 88.32 &pm; 1.57 | 87.23 &pm; 0.84 | 84.56 &pm; 0.88 | 78.76 &pm; 1.83 | 67.82 &pm; 1.92 |
> >
> > > What is the fundamental difference between the noises in the related literature and the one produced in the paper?
> >
> > A set of AR perturbations comprise a provably separable set of image vectors, and we use the manual-specification of CNN parameters to specify the function which separates them. Unlike other methods, AR perturbations are not optimized with a surrogate network and are resistant to strong data augmentations.
> >
> > [1] Theoretically Principled Trade-off between Robustness and Accuracy, ICML 2019 \
> > [2] Robustness May Be at Odds with Accuracy, ICLR 2019 \
> > [3] Better Safe Than Sorry: Preventing Delusive Adversaries with Adversarial Training, NeurIPS 2021

---

### Meta-Review · Area_Chair_MmyR · 2022-08-25

**Recommendation:** Accept
**Confidence:** Certain

**Metareview:**

The paper proposed a novel auto-regressive perturbation method to make the data unlearning. The method is independent to models and data, making it more easy to be used. Reviewers found the idea is novel and intuitively reasonable. The authors responded to reviewers' detailed questions about the method and experiments. The rebuttal succeeded to remove the confusions and convince us about the empirical significance. We suggest the authors improve the paper according to the review comments in the next version.

**Award:**

No

---

### Decision · Program_Chairs · 2022-09-14

Accept